# Genome-Wide Identification of *CCD* Gene Family in Six Cucurbitaceae Species and Its Expression Profiles in Melon

**DOI:** 10.3390/genes13020262

**Published:** 2022-01-28

**Authors:** Denghu Cheng, Zhongyuan Wang, Shiyu Li, Juan Zhao, Chunhua Wei, Yong Zhang

**Affiliations:** College of Horticulture, Northwest A&F University, Yangling, Xianyang 712100, China; denghucheng@nwafu.edu.cn (D.C.); zydx@nwafu.edu.cn (Z.W.); 15935411104@163.com (S.L.); zhaojuan125415@163.com (J.Z.); xjwend020405@nwafu.edu.cn (C.W.)

**Keywords:** carotenoids, *CCD* gene family, whole genome identification, expression analysis

## Abstract

The carotenoid cleavage dioxygenase (CCD) gene family in plants comprises two subfamilies: CCD and 9-cis-epoxycarotenoid dioxygenase (NCED). Genes in the *NCED* subfamily are mainly involved in plant responses to abiotic stresses such as salt, low temperature, and drought. Members of the *NCED* subfamily are the most important rate-limiting enzymes in the biosynthesis of abscisic acid (ABA). In the present study, genome-wide analysis was performed to identify *CCD* gene members in six Cucurbitaceae species, including watermelon (*Citrullus lanatus*), melon (*Cucumis melo*), cucumber (*C.sativus*), pumpkin (*Cucurbita moschata*), bottle gourd (*Lagenaria siceraria*), and wax gourd (*Benincasa hispida*). A total of 10, 9, 9, 13, 8, 8 *CCD* genes were identified in the six species, respectively, and these genes were unevenly distributed in different chromosomes. Phylogenetic analysis showed that *CCD* genes of the six species clustered into two subfamilies: *CCD* and *NCED*, with five and three independent clades, respectively. The number of exons ranged from 1 to 15, and the number of motifs were set to 15 at most. The cis-acting elements analysis showed that a lot of the cis-acting elements were implicated in stress and hormone response. Melon seedlings were treated with salt, low temperature, drought, and ABA, and then tissue-specific analysis of *CCDs* expression were performed on the root, stem, upper leaf, middle leaf, female flower, male flower, and tendril of melon. The results showed that genes in *CCD* family exhibited various expression patterns. Different *CCD* genes of melon showed different degrees of response to abiotic stress. This study presents a comprehensive analysis of *CCD* gene family in six species of Cucurbitaceae, providing a strong foundation for future studies on specific genes in this family.

## 1. Introduction

Carotenoid is a terpenoid that is a fat-soluble pigment widely distributed in nature. Carotenoid is present in a large number of plants and is responsible for the yellow, orange, and red colors of fruits and flowers. Carotenoid is an isopentene-like polymer containing 40 carbons. It has a conjugated double bond chain and absorbs light in the UV and blue light range [1,2]. The color of carotenoids varies with the number of conjugated double bonds [3]. Carotenoids are synthesized in all photosynthetic organisms (bacteria, algae, and plants) and some non-photosynthetic organisms, such as bacteria and fungi. Currently, more than 700 types of natural carotenoids have been characterized. Carotenoids can be divided into carotene and lutein according to the different chemical structures [4]. In plants, carotenoids mainly exist in the chloroplasts and colored body membranes, and form binding proteins with chlorophyll, which are responsible for the function of light absorption as auxiliary pigments in photosynthesis [5,6].

Tan et al. (1997) [7] identified for the first time the *NCED* gene *Vp14*, which catalyzes carotenoid decomposition in mutants of maize viviparous seeds through a transposon mutagenic technique. The gene regulates ABA biosynthesis by catalyzing the conversion of 9-cis-epoxide carotenoids into Xanthoxin, a precursor of ABA biosynthesis. The substrate catalyzed by Vp14 protein has an epoxide structure, thus *Vp14* is known as 9-cis-epoxycarotenoid dioxygenase (NCED) [7]. The discovery of *Vp14* was an instrumental step in the exploration of related genes in other species and organisms [8].

The *CCD* gene family comprises *CCD* and *NCED* subfamilies [9]. Nine members of the *CCD* family have been reported in *Arabidopsis*, four members of the *CCD* subfamily (*CCD1*, *4*, *7*, and *8*), and five members of the *NCED* subfamily (*NCED2*, *3*, *5*, *6*, and *9*). The *CCD* gene family is further divided into five subfamilies: *CCD1*, *CCD4*, *CCD7*, *CCD8*, and *NCED*, according to the differences in cleavage sites and substrates of the members in this family [10]. A new member of the *CCD* family, CCD-like (*CCDL*), was reported for the first time in tomato [11]. All CCD proteins contain the retinal pigment epithelium membrane protein (RPE65) domain, typical of enzymes involved in carotenoid cleavage [12]. Similarity between members of the CCD protein sequences is very low, with the exception of four highly conserved histidine residues and a small number of glutamate residues [13,14]. Although *CCDs* and *NCEDs* belong to the same gene family, their sequence homology is relatively low, and their activities and substrates are different [8,15].

Degradation of carotenoids in higher plants mainly occurs through enzymatic degradation and non-enzymatic degradation processes. Enzymatic degradation process comprises the hydroxylase pathway, the carotenoid cleavage dioxygenase (CCD) pathway, and the lipoxygenase pathway [16,17]. The non-enzymatic degradation process includes direct chemical oxidation degradation, photooxidation degradation, and thermal oxidation degradation. Enzymatic or non-enzymatic oxidative degradation of carotenoids results in biologically important carotenoid derivatives known as apocarotenoids. *CCDs* can specifically cleave the double bond of carotenoids and produce a variety of apocarotenoids, including pigments, flavor and aroma substances, and plant growth regulators, among which vitamin A and abscisic acid (ABA) are two common apocarotenoids [8,13,18,19]. The expression of *CCD* gene has an important relationship with the production of apocarotenoids. In previous studies, *LcNCED1* is highly expressed in roots, stems, and flowers of *Lycium barbarum*. The expression of *LcNCED1* in fruits was consistent with the accumulation of ABA [20]. LmCCD1 protein cleaved β-carotene and lycopene to produce β-ionone and pseudoionone, and the increase of *LmCCD1* expression was related to the increase of β-ionone content in lycium barbarum fruit [21]. Reduced *PhCCD1* transcription levels in petunias resulted in a 58–76% decrease in β-ionone synthesis [22]. The expression level of the *AtNCED* gene was highly consistent with the ABA biosynthesis in the root tip and reproductive organs [23].

At present, there are few reports on the functional characteristics of *CCD* gene in Cucurbitaceae.CCD4 protein was most closely related to carotenoid degradation in zucchini. The expression levels of *CpCCD4a* and *CpCCD4b* were higher in the varieties with the lowest carotenoid content, suggesting that *CCD4* gene played an important role in carotenoid cleavage [24]. Melon CmCCD1 protein cleaved phytoene to produce geranylacetone, lycopene to pseudoionone, and β-carotene to β-ionone. In addition, the expression of CmCCD1 protein was upregulated during the development of three different color melon varieties, which is crucial to the formation of aroma and flavor of melon fruits [25]. The regulation of gene transcription level is very important for the composition and accumulation of carotenoids in watermelon fruit. The high expression of *NCED* gene was one of the main reasons for the yellow and white color of watermelon flesh [26]. Bottle gourd and wild watermelon promoted the accumulation of lycopene in grafted watermelon fruits by down-regulating the expression of *NCED* and *CCD* genes [27].

## 2. Materials and Methods

### 2.1. Identification and Physicochemical Properties of CCD Gene Family Members in Cucurbitaceae

The protein sequence database of watermelon (*C. lanatus*, 97103, v2), melon (*C.*
*melo,* DHL92, v3.6.1), cucumber (*C. sativus*, Chinese Long, v3), pumpkin (*C. moschata,* Rifu, v1), bottle gourd (*L. siceraria*, USVL1VR-Ls, v1), and wax gourd (*B. hispida, B227*) were downloaded from the Cucurbitaceae genome database (http://cucurbitgenomics.org/, accessed on 1 December 2021). The following steps were used to identify CCD genes in the above six species. In the first step, protein sequences of all CCD genes in Arabidopsis thaliana were downloaded from the Uniprot database (https://www.uniprot.org/, accessed on 1 December 2021). Through TBtools software [28], the protein sequences of nine AtCCDs identified in Arabidopsis were searched by blast in the protein sequences database of the above six species, and the candidate sequences of CCDs protein sequences in the above six species were obtained. Step 2: The RPE65 (PF03055) domain sequences of all species were downloaded from the Pfam (http://pfam.xfam.org/, accessed on 1 December 2021) database. HMMER3.0 software was used to search the whole genome protein sequences of the above six species to obtain candidate CCDs protein sequences [29]. As the third step, the result of the combination of the first step and the second step, through the Conserved Domains Database (https://www.ncbi.nlm.nih.gov/cdd/?term=, accessed on 1 December 2021) and Pfam domain database (http://pfam.xfam.org/search#tabview=tab1, accessed on 1 December 2021) analysis of structure, all CCD gene family genes of the above six species were obtained by removing the sequence excluding the RPE65 domain. The ExPASy website (https://web.expasy.org/protparam/, accessed on 1 December 2021) was used to predict the CCDs protein amino acid quantity, isoelectric point, relative molecular mass, and grand average of hydropathicity. Using the BUSCA website (http://busca.biocomp.unibo.it/, accessed on 1 December 2021) and the WoLF PSORT website (https://wolfpsort.hgc.jp/, accessed on 1 December 2021) the CCDs protein subcellular localization was predicted. The location of the gene on the chromosome, the length of the gene, and the length of the gene CDS sequence are obtained from the Cucurbitaceae website.

### 2.2. Evolutionary Analysis

All the protein sequences of the CCDs in Cucurbitaceae were compared with the protein sequences of nine AtCCDs identified in Arabidopsis by using the program ClustalW, in the software MEGA7 (https://www.megasoftware.net/, accessed on 1 December 2021). The phylogenetic tree of the CCD gene family of Cucurbitaceae was constructed by MEGA7, using the neighbor-joining method (NJ) of the software, the self-help method of phylogeny test (Bootstrap method, Bootstrap = 1000), and partial deletion of gap or missing data. Finally, the phylogenetic tree of the Cucurbitaceae CCD gene family was constructed based on the Jones–Taylor–Thornton (JTT) model and rates among sites of Gamma Distributed (G).

### 2.3. Chromosomal Localization

The location information of the CCDs was obtained from the Cucurbitaceae website (http://cucurbitgenomics.org/, accessed on 1 December 2021), including the length of the chromosome, which chromosome it is located on, and the specific location on the chromosome. The picture showing the position of CCDs on the chromosome was obtained by using the TBtools software.

### 2.4. Gene Structure and Conserved Motifs Analysis

To investigate the gene structures of CCDs, the gff3 files of watermelon (*C. lanatus*, 97103, v2), melon (*C. melo*, DHL92, v3.6.1), cucumber (*C. sativus*, Chinese Long, v3), pumpkin (*C. moschata*, Rifu, v1), bottle gourd (*L. siceraria*, USVL1VR-Ls, v1), and wax gourd (*B. hispida*, B227) were subsequently downloaded from the Cucurbitaceae database (http://cucurbitgenomics.org/, accessed on 1 December 2021), and visualized by the software TBtools. The conserved motifs of all identified CCDs were analyzed and identified by the MEME website (https://meme-suite.org/meme/tools/meme, accessed on 1 December 2021) [30] and the number of conserved domains was set to 15 at most. By using the TBtools software, the results of CCDs structure and the conservative motifs analysis of CCDs protein can be obtained.

### 2.5. Cis-Acting Regulating Element Prediction

The TBtools software was used to extract the 2K sequence upstream of the translation initiation codon of Cucurbitaceae CCDs. The sequences were submitted to the PlantCare (http://bioinformatics.psb.ugent.be/webtools/plantcare/html/, accessed on 1 December 2021) website for cis-acting regulating element prediction.

### 2.6. Expression Analysis

In the current study, “TianJu” was used as the experimental material to analyze the *CCD* gene expression profile in melon in response to abiotic stress. “TianJu” is a melon variety developed by the Research Team of Melon Species Germplasm Resources and Genetic Breeding at the College of Horticulture, Northwest A & F University. The melon variety was planted in a light incubator and maintained under a light intensity of 300 μmol m^−2^ s^−^^1^, photoperiod of 16 h light/8 h dark, and a temperature cycle of 30 °C /22 °C until the melon seedlings grew to the three-leaf stage. Strong and consistent melon seedlings were selected for subsequent treatments. For salt treatment, roots were treated with 300 μmol/L NaCl, and leaves were then collected at 0 h, 1 h, 3 h, 6 h, 12 h, 24 h, and 48 h after treatment. The seedlings in the low-temperature group were maintained at 4 °C under a light intensity of 300 μmol m^−2^ s^−1^ and photoperiod of 16 h light/8 h dark. Leaves were collected at 0 h, 1 h, 3 h, 6 h, 12 h, 24 h, and 48 h after treatment. Drought was simulated in the drought-treated group by placing seedlings under a light incubator with a light intensity of 300 μmol m^−2^ s^−1^, photoperiod 16 h light/8 h dark, and temperature cycle of 30 °C/22 °C. The leaves in this group were collected at day 0, day 1, day 3, day 5, day 7, and day 9 after treatment. Re-watering (RW) treatment was performed on day 9, and leaves were collected on the day after watering treatment. ABA solution (100 μmol/L) was sprayed on the leaves of seedlings in the ABA group until the solution dripped from the leaves. The leaves in the ABA group were then collected and immediately placed in liquid nitrogen and transferred to the refrigerator at −80 °C for RNA extraction. The experimental material for tissue-specific expression analysis of the *CCD* gene family in melon was a “TianJu” variety planted in the D08 chamber of the scientific research greenhouse, South Campus of Northwest A & F University. The variety was allowed to grow until the flowering period to obtain female flowers under a suitable growth environment. The root, stem, upper leaf (leaf at the top of the plant), middle leaf (leaf at the middle of the plant), the female flowers that opened on the same day, the male flowers that opened on the same day, and tendrils were then harvested. The samples were placed on tin foil. Liquid nitrogen was added to the samples and then transferred to a refrigerator at −80 °C for subsequent extraction of RNA. Three biological replicates were set for each treatment in all the experiments, and three samples were used for each replicate.

Total RNA was extracted using the general plant total RNA extraction kit (Beijing Junrode Biotechnology Co., Ltd., Beijing, China). First chain cDNA was synthesized using the FastKing cDNA first chain synthesis kit (Tiangen Biochemical Technology Co., Ltd., Beijing, China) according to the manufacturer’s instructions. Specific primers for melon *CCDs* were designed using Primer 6.0 software. Primers were synthesized at Xi ’an Qingke Jersey Biotechnology Co., Ltd. (Xi ’an, China) Details on the primers are presented in Appendix A. Expression levels of *CCDs* in melon under abiotic stress and in different tissues were evaluated by Quantitative Real-time PCR instrument StepOnePlusTM (Applied Biosystems, Bedford, MA, USA). The total volume of the amplification system was 10 μL, comprising 5 μL SYBR, 3.6 μL ddH2O, 1 μL cDNA, and 0.2 μL upstream and downstream primers. The reaction program of Quantitative Real-time PCR analysis was set as follows: pre-denaturation at 94 °C for 3 min, denaturation at 94 °C for 10 s, annealing at 60 °C for 30 s and 40 cycles. Relative expression of genes was calculated by the 2^−ΔΔCT^ method, and actin was used as an internal control [31].

## 3. Results

### 3.1. Genome-Wide Identification of CCD Gene Family in Cucurbitaceae

A total of 57 CCDs protein sequences were identified from six Cucurbitaceae species through genome-wide analysis using TBtools and HMMER 3.0 tools. Sequences without RPE65 domain were eliminated through analysis using the Conserved Domains Database and the Pfam Database. The results showed 10, 9, 9, 13, 8 and 8 CCDs protein sequences in watermelon (*C. lanatus*), melon (*C. melo*), cucumber (*C. sativus*), pumpkin (*C. moschata*), bottle gourd (*L. siceraria*), and wax gourd (*B. hispida*), respectively (Table 1). These genes were subsequently named according to their homologous relationship with *CCDs* in *Arabidopsis*. The nucleotide sequences of *CCDs* showed significant variations; however, they encoded proteins with a length ranging from 500 to 600 amino acids and molecular weights ranging between 60 kD and 70 kD. Notably, the GRAVY (grand average of hydropathicity) values of all identified CCDs proteins were less than 0, implying that the proteins were hydrophilic. Moreover, most of the CCDs proteins had predicted isoelectric point (PI) values below seven, and were mainly localized in chloroplasts.

### 3.2. Evolutionary Analysis of CCD Gene Family in Cucurbitaceae

All the 57 CCD proteins from Cucurbitaceae were used to construct a phylogenetic tree using MEGA7 software to further explore the evolution of the *CCD* gene family (Figure 1), with nine *Arabidopsis* CCDs protein sequences as reference sequences. Analysis of the phylogenetic tree showed that the CDDs protein was clustered into two subfamilies (CCD and NCED). The CCD and NCED subfamilies comprised five (CCD1, CCD4, CCD7, CCD8, and CCDL) and three (NCED3, NCED5, and NCED6) clades, respectively. These findings were consistent with findings reported in a previous study on tobacco [1]. The findings showed that only watermelon had two proteins in the CCDL clade, whereas other Cucurbitaceae species only had one gene in the CCDL clade. Pumpkin and wax gourd had three and one proteins in the NCED5 clade, respectively, whereas all other species had two members in the NCED5 clade. Cucurbitaceae species only had one protein in CCD1, CCD4, CCD7, and CCD8 subfamilies, apart from pumpkin. Bottle gourd had no protein in the CCDL clade; however, bottle gourd CCDs proteins were observed in other clades like other species of Cucurbitaceae. The genetic distance between CCD8 and NCED proteins was relatively large, whereas the distance between CCD4 and NCED proteins was relatively small.

### 3.3. Members of CCDs Showed Variations in Chromosomal Localization in Cucurbitaceae

Distribution maps of the 57 *CCDs* identified on chromosomes were generated using TBtools software (Figure 2). The results showed that Cucurbitaceae *CCDs* are unevenly distributed on the chromosomes of different species. *CCDs* were localized on five chromosomes in bottle gourd and wax gourd, seven chromosomes in watermelon, six chromosomes in cucumber, and nine chromosomes in melon and pumpkin. The findings showed presence of two *CCDs* on chromosome 6, 7, and 11 of watermelon, chromosome 2 and 4 of cucumber, chromosome 4 and 16 of pumpkin, chromosome 7 and 9 of bottle gourd, and chromosome 1, 9, and 12 of wax gourd. Three *CCDs* were localized on chromosome 17 of pumpkin, whereas one *CCD* gene was observed in each chromosome of melon. Notably, *CCDs* were arranged in tandem on their chromosomes in melon and the other five species.

### 3.4. CCD Gene Family Presents Characteristic Gene Structure and Conserved Motifs in Cucurbitaceae

Conserved motifs in CCD proteins were explored using MEME webserver (https://meme-suite.org/meme/tools/meme, accessed on 1 December 2021). Structural characteristics of CCDs comprising the coding DNA sequence (CDS), untranslated region (UTR), and intron were analyzed using TBtools software. The results showed that the number of motifs in the *NCED* subfamily was significantly higher than that in the *CCD* subfamily (Figure 3). The difference in number and species of conserved domains among different clades of *CCD* subfamily was significantly higher than that among different clades of NCED subfamily. Notably, LsiCCD1 protein lacked motif 14, and LsiCCD8 protein lacked motif 5 and motif 6, whereas all other CCDs proteins had motif 5, motif 6, motif 7, and motif 14. These findings indicate that these four motifs are highly conserved in the *CCD* gene family and may play an essential role in some common functions. The CCD proteins with close evolutionary relationships as shown in the phylogenetic tree exhibited similar motifs composition (Figure 3). The number of motifs in CCD1 and CCD4 clades was significantly higher relative to the number of motifs in other clades. AtCCD7 protein in CCD7 clade had a unique motif 10, but lacked motif 2, which was present in all other members in this clade. Analysis of the CCD-like clade showed that only BhCCDL protein had motif 4, and did not have motif 2, present in all other members of this clade. CmoCCD1 protein in CCD1 clade had motif 13, which was absent in other clades. Analysis of CCD4 clade showed that all members had the same motifs. Notably, motif 6 was only observed in AtCCD4 protein, and motif 3 was only present in LsiCCD4 protein. Analysis of the three clades of the NCED subfamily showed that all members had all 15 identified motifs. The number, type, and order of motifs were similar across different clades.

The findings showed significant differences in gene structure among different members of the *CCDs*. The members of this family showed a minimum of one exon and a maximum of 15 exons, and a minimum of zero introns, and a maximum of 14 introns. Further analysis of the gene structure showed that the gene structure of the *CCD* subfamily was more complex compared with the gene structure of the *NCED* subfamily. Analysis of the different clades of *CCD* subfamily showed that the gene structure of CCD4 clade was different from that of other clades in that each gene of the CCD4 clade had no introns. Each member of CCD8 clade had six exons. Analysis of CCD7 clade showed the presence of six exons in *AtCCD7* gene, eight exons in the *CmCCD7* gene, and seven exons in other members of this clade. CCDL clade and CCD1 clade members showed the significantly highest number of exons compared with the number of exons in other clades. The results showed that all members of the *NCED* subfamily had no introns, except *CmNCED5b* and *CmoNCED3b* genes, which had only one intron. Notably, the number of introns in the *NCED* subfamily was significantly lower, relative to that in the *CCD* subfamily.

### 3.5. Cis-Acting Elements of CCDs Were Identified in Cucurbitaceae

*Cis*-acting elements of the 2K sequence upstream of the CDS of all Cucurbitaceae *CCDs* were explored using the PlantCARE website. The *CCD* gene family mainly encodes hormones and proteins implicated in the stress response [32], and thus eight related elements were selected for analysis in the present study (Figure 4). The elements included the salicylic acid-responsive element, the abscisic acid-responsive element, the gibberellin-responsive element, the MeJA-responsive element, the defense and stress-responsive element, the drought-induced responsive element, the auxin-responsive element, and the low-temperature responsive element. The results showed differences in the number of response elements among the different subfamilies. The number of drought response elements in the *CCDs* was the least (19), whereas the number of abscisic acid response elements was the highest (148), indicating that most members of the *CCDs* were implicated in the ABA response.

### 3.6. Expression Profiles of CCD Candidate Genes in Different Tissues of Melon

The qRT-PCR results of *CCDs* in different tissues of melon showed differences in expression levels of different *CCD* candidate genes in different tissues (Figure 5). The expression level of *CmCCD7* in roots was higher compared with the expression level in other tissues. The expression levels of *CmCCD1* and *CmCCD4* were lower in roots compared with the expression levels in other tissues. Four genes showed relatively high expression levels in the stem, including *CmCCD1, CmCCD4, CmNCED3*, and *CmNCED6*. Notably, seven genes showed low relative expression levels in the upper leaves, and only *CmCCD4* showed a significantly high expression level. Three genes (*CmCCD1, CmCCD4,* and *CmNCED5a*) were highly expressed in middle leaves, and *CmCCD4* was significantly overexpressed in upper leaves. *CmCCD7* was the only gene that showed significantly low expression levels in female flowers, whereas all other genes showed high expression levels. The relative expression levels of six genes (*CmCCD1, CmCCD4, CmCCD8, CmCCDL, CmNCED5a,* and *CmNCED5b*) were high in male flowers, with *CmNCED5b* having the highest relative expression level. Five genes showed high expression, and four genes showed low expression levels in tendrils.

### 3.7. Expression Profiles of CCD Family Candidate Genes in Response to Abiotic Stress in Melon

The qRT-PCR results of samples obtained under abiotic stress showed significant differences in gene expression levels of melon *CCDs* after treatment with salt, cold (4 °C), drought, and ABA (Figure 6 and Figure 7). Expression of *CmCCD4* and *CmCCD8* was downregulated, whereas expression of *CmNCED3* was significantly upregulated after salt treatment. Expression levels of *CmNCED5a* and *CmNCED5b* increased at the start of the experiment then decreased, and the expression level of *CmNCED5b* showed double peaks and exhibited significant upregulation at 3 h. Expression levels of *CmCCD7* and *CmNCED6* decreased at the beginning of the treatment then increased, and the expression profile showed double peaks. Expression levels of *CmCCD1* and *CmCCD4* increased at the start of treatment at 4 °C then decreased, and the expression level of *CmCCD4* significantly increased at 3 h. Expression levels of *CmNCED3* and *CmNCED6* showed an increase at the start of low-temperature treatment, then a decrease and a further increase, and were significantly upregulated at 24 h and 3 h, respectively. Expression levels of *CmCCD7, CmCCD8, CmNCED5a,* and *CmNCED5b* decreased at the start of low-temperature treatment, and then increased in later stages of the experiment. Expression of *CmCCD7, CmCCD8, CmNCED5a,* and *CmNCED5b* was significantly downregulated at 1 h, 48 h, 48 h, and 1 h, respectively. Expression of *CmCCD1* gene was upregulated, whereas expression of *CmNCED5a* gene was downregulated throughout the drought treatment experiment, with significant downregulation of *CmNCED5a* expression observed on day 9. The gene expression levels of *CmCCD4*, *CmCCD7*, *CmCCD8* and *CmNCED6* were downregulated only on the first day, and upregulated at other times. Expression levels of *CmNCED3* and *CmNCED5b* initially decreased during the drought treatment, increased and then further decreased. Expression of *CmNCED3* was significantly upregulated on day 3. Expression of most genes showed downregulation after RW treatment. Expression of *CmCCD1* was upregulated after ABA treatment, with significant upregulation observed at 24 h. ABA treatment downregulated expression of *CmNCED5a* with significant downregulation observed from 6 h to the end of the experiment. *CmNCED3* gene expression was upregulated at most time points, with downregulation observed only at 48 h, and significant upregulation of *CmNCED3* expression was observed from 3 h to the end of the ABA treatment experiment. *CmCCD4, CmCCD7 CmNCED5b,* and *CmNCED6* expression levels showed a decrease, then an increase during ABA treatment. Expression of *CmCCD7* showed significant upregulation at 24 h during ABA treatment.

## 4. Discussion

The *CCD* gene family belongs to one of the smaller gene families in plants, owing to the low number of genes in the family [33]. In the present study, 10, 9, 9, 13, 8, and 8 CCDs protein sequences were identified in watermelon, melon, cucumber, pumpkin, bottle gourd, and wax gourd, respectively (Table 1). A total of 30, 19, 7, and 11 CCDs protein sequences were identified in oilseed rape [34], tobacco [1], tomato [11], and pepper [12], respectively. Molecular characteristics of all CCDs protein members of Cucurbitaceae are different, and proteins in the same clade showed different molecular weights and isoelectric points. Prediction results showed that CCDs proteins were distributed in the chloroplast, mitochondria, cytoplasm, and plasma membrane. Notably, more than half of the proteins were localized in the chloroplast, implying that CCDs proteins play an important role in the function of the chloroplast (Table 1). 

The phylogenetic tree showed that the CCDs proteins in Cucurbitaceae clustered into two subfamilies (CCDs and NCEDs) and eight clades (CCD1, 4, 7, 8 and L; and NCED, 3, 5, 6). NCED2 and NCED9 clades were not identified, owing to the lack of corresponding homologues from *Arabidopsis*. Melon, cucumber, pumpkin, and wax gourd each had one gene in the CCDL clade first reported in tomato [11], whereas watermelon and bottle gourd had two and zero members in this clade, respectively. The NCED subfamily was more closely related to CCDL and CCD4 clades, and distantly related to CCD8 clade. In addition, the findings showed that the NCED subfamily evolved from the CCD4 clade (Figure 1).

Previous studies report that the position of exons and introns in gene family members plays an important role in evolution [35]. In the present study, motif and gene structure analysis showed that the number and location of exons and introns of motifs and genes in the same population were similar, which was consistent with the topology of phylogenetic trees. Further, motif and gene structure results showed that the number and location of motifs and exons and introns in the same clade were similar, consistent with the topology of the phylogenetic trees. All members of the NCED clade except *CmNCED5b* and *CmoNCED3b* had no introns (Figure 3). However, the motifs of members in the *NCED* subfamily were more conserved compared with those of the *CCD* subfamily, which is a common feature in plants [1]. 

The promoter located upstream of the initiation codon region is an important *cis*-acting element for genes and an important locus for transcription modulation [36]. Many *cis*-acting elements of the *CCDs* were predicted in *Cucurbitaceae* to be related to hormone and stress responses (Figure 4). These findings indicate that expression of *CCDs* are regulated by different *cis*-acting elements in hormone and abiotic stress-related promoters during the growth and development of Cucurbitaceae plants. In the present study, the expression levels of *CCDs* in melon exposed to ABA and four abiotic stresses were explored.

The *CCD* gene family is an ancient gene family in plants that plays a vital role in plant growth and stress response [34]. CCDs proteins are implicated in the production of various hormones and aromatic substances in plants [37]. *CCD1* is the most widely studied gene in the *CCD* gene family. CCD1 protein is mainly responsible for the cleavage of carotenoids to form aromatic substances in flowers and fruits [38]. CCD1 protein is involved in the formation of β-ziroxone, pseudoionone, and geranylacetone flavor volatile compounds in tomato [39]. Further, CCD1 protein regulates the release of β-zionone in morning glens [22]. The results of the present study showed that *CmCCD1* was highly expressed in stems and tendrils, but only low expression levels were observed in roots and the upper leaf (Figure 5). This finding was different from the expression patterns of *SlCCD1a*, and *SlCCD1b* reported in tomato [11], indicating that *CCD1* exhibits different expression patterns in different plants. Salt, drought, and ABA upregulated expression of *CmCCD1*. The cleavage products of cleavage of carotenoids by the CCD4 protein are related to the coloration of plant flesh and flower organs, and are also implicated in the production of aromatic substances [40,41,42]. CCD4 protein controls the color of peach fruit. Expression of the CCD4 protein leads to carotenoid degradation in white flesh genotype peaches, and the yellow color of peaches results from the inactivation of CCD4 protein [43]. The lily tenth segment changes from yellow to white because the lily CCD4 protein cleaves the carotenoids present in lily [44]. The change of chrysanthemum color from white to yellow was due to the increase of carotenoid content caused by the inactivation of CCD4 protein [45]. This suggests that we can use CRISPR/Cas9 and RNAi techniques to control the expression of CCD protein identified in this study, so as to change plant color, plant type, and other traits, and improve the nutritional quality of fruits. In the current study, *CmCCD4* was highly expressed in stems and leaves (Figure 5). Previous studies reported that *MdCCD4* is highly expressed in apple leaves [46], indicating that *CCD4* may play an important role in the growth and development of leaves. Moreover, the *CmCCD4* was highly sensitive to drought stress (Figure 6), which was consistent with previous findings that the *GHCCD4a-A* is upregulated in upland cotton under drought stress [47]. CCD7 and CCD8 are two key proteins in the plant hormone witcholactone synthesis pathway, and their role may be catalyzing carotenoid degradation [48]. Downregulation of CCD8 protein expression in kiwifruit is associated with increased branch development and delayed leaf senescence [49]. CCD7 protein is implicated in the production of witcholactone, branching, and arbuscular mycorrhizal-induced apopolyboid production in tomato [50]. In addition, CCD7 protein is the control protein of EMS mutagenesis in rice, leading to rice dwarfism and tillering increase [51]. A previous study used CRISPR/Cas9 technology to knock out *CCD8* in tobacco, and the findings showed an increase in the branches of tobacco and a decrease in plant height [52]. Furthermore, CCD8 protein plays the role of strigolactone in bud branching, mycorrhizal symbiosis, and parasitic weed interactions in peas [53]. In the present study, expression of *CmCCD7* was tissue specific, with a higher expression level in roots (Figure 5). Notably, previous findings indicate that the expression level of *NtCCD7* in roots of tobacco was higher relative to the expression level in other tissues [1], implying that *CCD7* plays an important role in root growth and development. Drought induced sustained upregulation of *CmCCD7* and *CmCCD8* expression, indicating that these two genes play an important role in response to drought stress (Figure 6). 

Abscisic acid is an important plant hormone implicated in the modulation of various plant development processes and plays an important regulatory role in different types of stresses [54]. Members of the *NCED* subfamily proteins are rate-limiting enzymes in the biosynthesis of ABA hormone [55]. Increase in ABA content in plants under stress is highly correlated with an increase in the expression of genes in the *NCED* subfamily [56], implying that *NCED* genes play an important role in plant response to stress. Under drought stress, the expression of *NCED* in roots, stems, and leaves of watermelon seedlings was up-regulated, suggesting that NCED proteins may regulate the synthesis and decomposition of endogenous ABA [57]. Under salt stress, the content of abscisic acid in cucumber was reduced by 67.68% by applying putrescine, but the expression of *NCED* gene was up-regulated and the content of ABA was increased by applying the inhibitor [58]. *TaNCED* expression in wheat is significantly upregulated by drought stress, whereas overexpression of *TaNCED* in *Arabidopsis* promotes an increase in endogenous ABA content, enhanced *Arabidopsis* tolerance to drought stress, and delayed seed germination [59]. PpNCED1 and VVNCED1 proteins in peach initiated ABA biosynthesis at the early stage of fruit ripening, and ABA accumulation plays a key role in the regulation of ripening and senescence in peach and grape fruits [60]. Overexpression of *PvNCED1* of kidney beans in wild tobacco increases the ABA content, and significantly improves tolerance of the tobacco plants to drought stress [61]. Upregulation of *CstNCED* expression is observed when saffron stigma is subjected to low temperature, water, salt, exogenous ABA, and other biological stresses [62]. A high expression level of *AtNCED3* in *Arabidopsis* roots promotes the synthesis of ABA and regulates the growth of *Arabidopsis* lateral roots [23]. Overexpression of *Arabidopsis AtNCED3* lead to increased endogenous ABA level and increased drought tolerance in these plants [63]. *AtNCED3* was the main response gene in the leaves of *Arabidopsis* under stress [23]. In the present study, the expression level of *CmNCED3* was higher in stems, female flowers, and tendrils, but lower in leaves (Figure 5). Moreover, *CmNCED3* expression was upregulated in melon leaves under all abiotic stress, indicating that *CmNCED3* plays an important role in stress response (Figure 7). Further studies should explore the role of *CmNCED3* of melon in depth and conduct gene cloning to elucidate the function of the *NCED3* gene. AtNCED5 protein regulated ABA synthesis in *Arabidopsis* seeds, thus regulating seed maturation and dormancy [23]. In the current study, expression levels of *CmNCED5a* and *CmNCED5b* in female and male flowers were significantly higher relative to the expression levels in other tissues (Figure 5). This finding was consistent with the expression pattern of *NtNCED5a* in tobacco [1], implying that NCED5 protein plays an important role in flower growth and development. Abiotic stress treatments used in the present study all downregulated the expression of *CmNCED5a* (Figure 7) to varying degrees. *CmNCED6* was highly expressed in stems, female flowers, and tendrils, and its expression was upregulated by drought stress (Figure 5 and Figure 7). In summary, the results of the present study provide a basis for further studies on the role of *CCDs* in Cucurbitaceae. However, further studies should be conducted to validate their functions.

## 5. Conclusions

The apocarotenoids produced by the cleavage of CCD subfamily proteins play an important role in influencing crop flavor and plant morphogenesis. The NCED subfamily proteins are the most important rate-limiting enzymes in the biosynthesis of plant hormone ABA, and are widely involved in the response process of plants to abiotic stress. Therefore, it is of great significance to study the *CCDs*. However, there are few reports about the function of *CCDs* in Cucurbitaceae. This study is the first to identify and bioinformatics analysis *CCD* genes in six species of Cucurbitaceae. In the present study, ten CCDs protein sequences were identified in watermelon, nine in melon, nine in cucumber, thirteen in pumpkin, eight in bottle, and eight in wax gourd. Notably, these *CCDs* were distributed on different chromosomes. Phylogenetic analysis revealed eight clades based on the homologous relationship with *Arabidopsis*. Gene structure and conserved motif analysis showed that members of the same clade are highly conserved, consistent with the results on evolutionary relationships. Furthermore, the findings of this study indicate that the promoters of the six species *CCDs* in Cucurbitaceae comprise several potential *cis*-acting elements involved in hormonal and physiological stress. Additionally, the expression patterns of *CCDs* in melon under different tissues, ABA, and abiotic stresses were explored in this study. Comparative analysis showed that some melon *CCDs* are implicated in important roles in plants; however, further research should be conducted to verify their functions. For example, *CmNCED5a* and *CmNCED5b* were upregulated in female and male flowers, indicating that they play essential roles in the growth and development of flower organs. *CmNCED3* and its homologous gene, *AtNCED3,* were highly expressed under abiotic stress, implying that *CmNCED3* plays a vital role in environmental stress responses. This study provides a solid foundation for further research on the function of the *CCDs* in Cucurbitaceae and the interactions between the members of this gene family.

## Figures and Tables

**Figure 1 genes-13-00262-f001:**
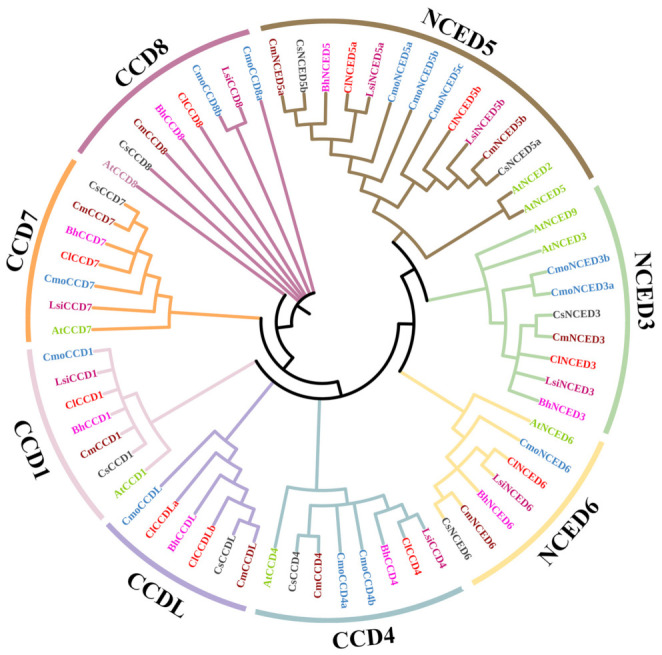
Evolutionary analysis of CCD proteins in Cucurbitaceae. The phylogenetic tree of the *CCD* gene family of Cucurbitaceae was constructed by MEGA7, using the neighbor-joining method (NJ) of the software. Phylogenetic analysis showed that CCD proteins of the six species clustered into two subfamilies: CCD and NCED, with five and three independent clades, respectively. Numbers on nodes represent bootstrap values.

**Figure 2 genes-13-00262-f002:**
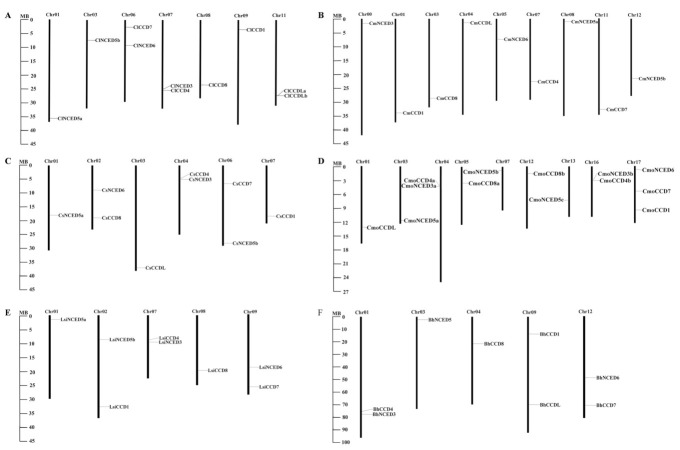
Chromosomal distribution of *CCDs* in Cucurbitaceae. The chromosome number is labelled on top of each chromosome. The left scale represents the length of chromosomes, and scale is expressed in megabase (Mb). Note: (**A**) *C. lanatus*; (**B**) *C. melo*; (**C**) *C. sativus*; (**D**) *C. moschata*; (**E**) *L. siceraria*; (**F**) *B. hispida*.

**Figure 3 genes-13-00262-f003:**
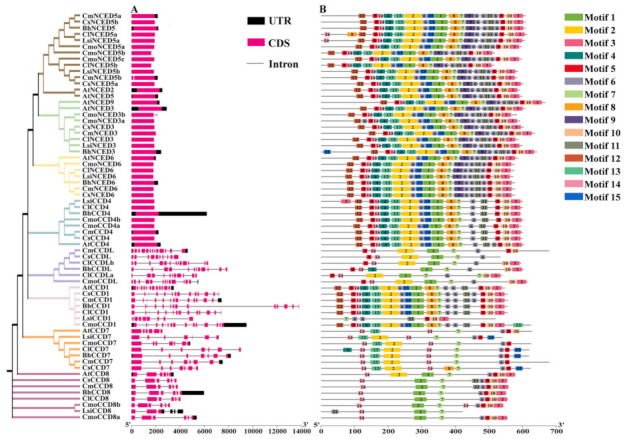
Gene structure of *CCDs* and conserved motifs analysis of CCD proteins in Cucurbitaceae based on phylogenetic relationships. (**A**) Gene structure of Cucurbitaceae *CCDs*. Black boxes indicate the untranslated 5′-3′ regions; red boxes indicate CDS; black lines represent introns. (**B**) The conserved motifs of all identified genes were analyzed and identified by the MEME website and the number of conserved domains was set to 15 at most.

**Figure 4 genes-13-00262-f004:**
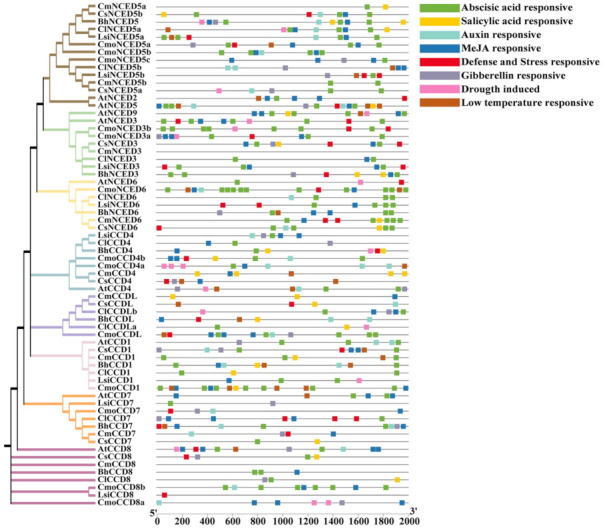
Cucurbitaceae *CCDs* promoter *cis*-acting elements analysis based on their phylogenetic relationships. *Cis*-acting elements of the 2K sequence upstream of the CDS of all Cucurbitaceae *CCDs* were explored using the PlantCARE website and different color indicated different *cis*-acting elements.

**Figure 5 genes-13-00262-f005:**
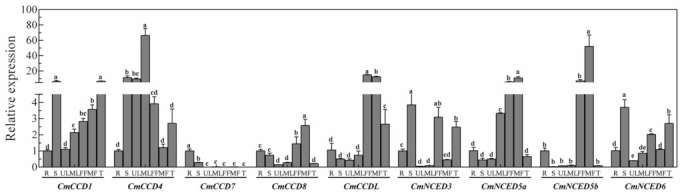
qRT-PCR analysis of *CCDs* and *NCEDs* in different tissues. Relative expression of genes was calculated by the 2^−ΔΔCT^ method. Different letters represent statistically significant differences (ANOVA with Turkey post-hoc analysis, 5% level). The mean ± S.D. of the biological replicates is presented. Note: R: Root; S: Stem; UL: Upper leaf; ML: Middle leaf; FF: Female flowers; MF: Male flowers; T: Tendril.

**Figure 6 genes-13-00262-f006:**
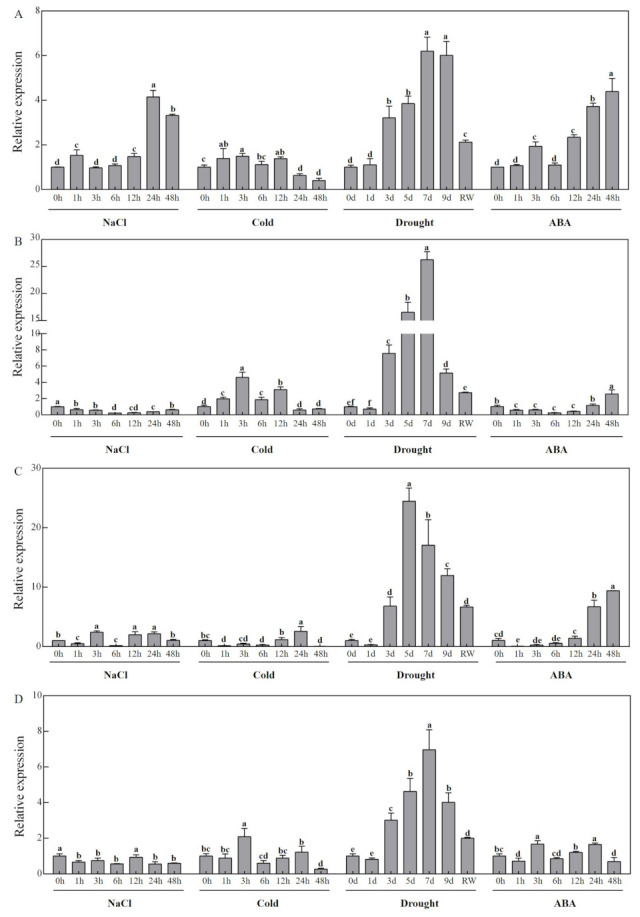
qRT-PCR analysis of *CCDs* genes in response to abiotic stress in melon ((**A**) *CmCCD1*; (**B**) *CmCCD4*; (**C**) *CmCCD7*; (**D**) *CmCCD8*). Relative expression of genes was calculated by the 2^−ΔΔCT^ method. Different letters represent statistically significant differences (ANOVA with Turkey post-hoc analysis, 5% level). The mean ± S.D. of the biological replicates is presented.

**Figure 7 genes-13-00262-f007:**
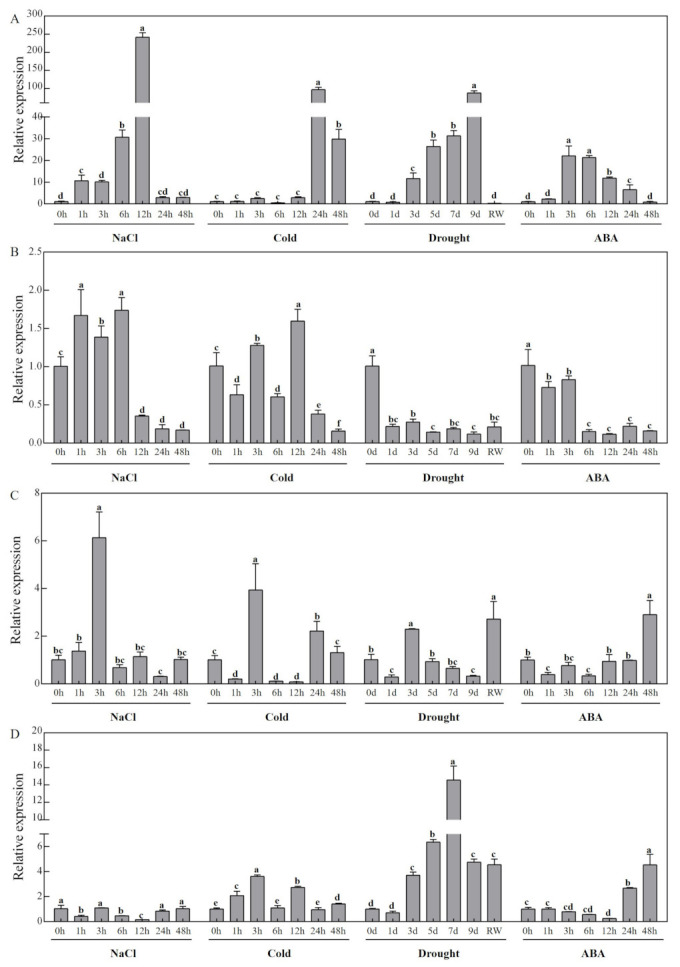
qRT-PCR analysis of *NCED* genes in response to abiotic stress in melon ((**A**) *CmNCED3*; (**B**) *CmNCED5a*; (**C**) *CmNCED5b*; (**D**) *CmNCED6*). Relative expression of genes was calculated by the 2^−ΔΔCT^ method. Different letters represent statistically significant differences (ANOVA with Turkey post-hoc analysis, 5% level). The mean ± S.D. of the biological replicates is presented.

**Table 1 genes-13-00262-t001:** The genes and encoded protein properties of all *CCDs* identified in this study.

Species	Gene Name	Gene ID	PI	Molecular Weight (kDa)	Chromosome Location	Gene Length (bp)	CDS Length (bp)	Number of Amino Acids (aa)	Grand Average of Hydropathicity	Subcellular Localization
*Citrullus lanatus*	*ClCCD1*	Cla97C09G166380	6.21	61.47	Chr09: 3477850 .. 3484797 (+)	6948	1644	547	−0.291	Cytoplasm
	*ClCCD4*	Cla97C07G137780	6.00	65.07	Chr07: 25368259 .. 25370034 (+)	1776	1776	591	−0.234	Chloroplast
	*ClCCD7*	Cla97C06G111770	7.04	68.63	Chr06: 2532539 .. 2541011 (−)	8473	1833	610	−0.215	Chloroplast
	*ClCCD8*	Cla97C08G155420	7.91	61.00	Chr08: 23384307 .. 23388101 (−)	3795	1644	547	−0.235	Chloroplast
	*ClCCDLa*	Cla97C11G221170	6.30	60.37	Chr11: 27257473 .. 27262531 (+)	5059	1596	531	−0.301	Cytoplasm
	*ClCCDLb*	Cla97C11G221180	6.84	67.14	Chr11: 27264241 .. 27270176 (+)	5936	1764	587	−0.404	Chloroplast
	*ClNCED3*	Cla97C07G137260	5.81	64.04	Chr07: 24786951 .. 24788687 (+)	1737	1737	578	−0.243	Chloroplast
	*ClNCED5a*	Cla97C01G024630	6.28	67.74	Chr01: 35670071 .. 35671885 (+)	1815	1815	604	−0.404	Chloroplast
	*ClNCED5b*	Cla97C03G058030	5.61	56.93	Chr03: 7238075 .. 7239598 (−)	1524	1524	507	−0.28	Cytoplasm
	*ClNCED6*	Cla97C06G117340	7.3	62.69	Chr06: 9148622 .. 9150322 (+)	1701	1701	566	−0.279	Cytoplasm
*Cucumis melo*	*CmCCD1*	MELO3C023555.2	6.01	61.28	Chr01: 33650150 .. 33657132 (+)	6983	1644	547	−0.278	Cytoplasm
	*CmCCD4*	MELO3C016224.2	6.1	65.08	Chr07: 22480061 .. 22482158 (−)	2098	1779	592	−0.235	Chloroplast
	*CmCCD7*	MELO3C022291.2	7.82	75.47	Chr11: 32532564 .. 32539629 (+)	7066	2010	669	−0.422	Chloroplast
	*CmCCD8*	MELO3C011142.2	6.13	61.14	Chr03: 28383115 .. 28386694 (+)	3580	1653	550	−0.232	Chloroplast
	*CmCCDL*	MELO3C003462.2	7.12	76.52	Chr04: 1417853 .. 1422210 (−)	4358	2010	669	−0.404	Chloroplast
	*CmNCED3*	MELO3C027057.2	6.06	69.64	Chr00: 1582539 .. 1584421 (+)	1883	1881	626	−0.214	Chloroplast
	*CmNCED5a*	MELO3C007127.2	8.34	67.09	Chr08: 991369 .. 993401 (−)	2033	1800	599	−0.389	Chloroplast
	*CmNCED5b*	MELO3C002744.2	6.32	65.23	Chr12: 21375227 .. 21377259 (−)	2033	1728	575	−0.378	Mitochondrion
	*CmNCED6*	MELO3C023086.2	7.98	63.02	Chr05: 7293775 .. 7295481 (−)	1707	1707	568	−0.278	Mitochondrion
*Cucumis sativus*	*CsCCD1*	CsaV3_7G031010	5.93	61.34	Chr07: 19638561 .. 19646579 (+)	7679	1644	547	−0.276	Cytoplasm
	*CsCCD4*	CsaV3_4G007180	5.83	65.04	Chr04: 4915379 .. 4917916 (−)	2538	1776	591	−0.236	Chloroplast
	*CsCCD7*	CsaV3_6G008730	6.73	69.42	Chr06: 7029451 .. 7035039 (+)	5589	1845	614	−0.345	Chloroplast
	*CsCCD8*	CsaV3_2G030870	6.17	60.76	Chr02: 20280011 .. 20284196 (−)	4186	1644	547	−0.235	Chloroplast
	*CsCCDL*	CsaV3_3G048570	8.92	59.86	Chr03: 39663250 .. 39666931 (+)	3682	1578	525	−0.364	Chloroplast
	*CsNCED3*	CsaV3_4G007760	6.04	65.82	Chr04: 5381592 .. 5384221 (−)	2630	1785	594	−0.273	Chloroplast
	*CsNCED5a*	CsaV3_1G032340	6.82	66.92	Chr01: 19294926 .. 19297672 (−)	2747	1782	593	−0.335	Chloroplast
	*CsNCED5b*	CsaV3_6G051790	8.34	67.15	Chr06: 30131435 .. 30133535 (+)	2101	1803	600	−0.377	Chloroplast
	*CsNCED6*	CsaV3_2G012080	8.78	63.02	Chr02: 9534360 .. 9536321 (+)	1962	1707	568	−0.298	Chloroplast
*Cucurbita moschata*	*CmoCCD1*	CmoCh17G009320	6.82	69.02	Chr17: 8291575 .. 8300490 (−)	8916	1848	615	−0.186	Cytoplasm
	*CmoCCD4a*	CmoCh04G006500	5.7	64.61	Chr04: 3223403 .. 3225163 (−)	1761	1761	586	−0.222	Chloroplast
	*CmoCCD4b*	CmoCh16G005460	6.05	65.30	Chr16: 2645574 .. 2647358 (+)	1785	1785	594	−0.231	Chloroplast
	*CmoCCD7*	CmoCh17G005060	6.45	68.79	Chr17: 4737408 .. 4741956 (−)	4549	1848	615	−0.258	Chloroplast
	*CmoCCD8a*	CmoCh05G006180	8.73	60.84	Chr05: 3082399 .. 3087453 (−)	5055	1644	547	−0.221	Chloroplast
	*CmoCCD8b*	CmoCh12G002040	6.58	59.53	Chr12: 1340192 .. 1343166 (−)	2975	1608	535	−0.23	Chloroplast
	*CmoCCDL*	CmoCh01G014860	6.59	68.36	Chr01: 11513752 .. 11518925 (−)	5174	1821	606	−0.293	Chloroplast
	*CmoNCED3a*	CmoCh04G006910	6.49	64.70	Chr04: 3434234 .. 3435985 (−)	1752	1752	583	−0.254	Mitochondrion
	*CmoNCED3b*	CmoCh16G004950	5.61	63.95	Chr16: 2388840 .. 2390824 (+)	1985	1725	574	−0.291	Chloroplast
	*CmoNCED5a*	CmoCh03G013970	6.58	65.79	Chr03: 10201389 .. 10203155 (+)	1767	1767	588	−0.389	Chloroplast
	*CmoNCED5b*	CmoCh07G001020	5.86	57.28	Chr07: 595470 .. 596999 (−)	1530	1530	509	−0.314	Cytoplasm
	*CmoNCED5C*	CmoCh13G005570	6.06	65.71	Chr13: 6389844 .. 6391610 (−)	1767	1767	588	−0.287	Chloroplast
	*CmoNCED6*	CmoCh17G001180	7.26	62.53	Chr17: 638459 .. 640162 (−)	1704	1704	567	−0.226	Chloroplast
*Lagenaria siceraria*	*LsiCCD1*	Lsi02G025710	6.33	58.89	Chr02: 32164396 .. 32169161 (+)	4766	1536	511	0.037	Plasma membrane
	*LsiCCD4*	Lsi07G007810	5.91	65.09	Chr07: 8595598 .. 8597373 (−)	1776	1776	591	−0.207	Chloroplast
	*LsiCCD7*	Lsi09G016990	6.3	59.89	Chr09: 25505283 .. 25512032 (+)	6750	1596	531	−0.212	Cytoplasm
	*LsiCCD8*	Lsi08G010850	8.34	46.52	Chr08: 19415522 .. 19419519 (−)	3998	1245	414	−0.161	Chloroplast
	*LsiNCED3*	Lsi07G008450	5.94	65.41	Chr07: 9468950 .. 9470722 (−)	1773	1773	590	−0.265	Chloroplast
	*LsiNCED5a*	Lsi01G001400	6.65	66.92	Chr01: 1451654 .. 1453447 (−)	1794	1794	597	−0.385	Chloroplast
	*LsiNCED5b*	Lsi02G008960	6.22	65.05	Chr02: 8564357 .. 8566090 (+)	1734	1734	577	−0.325	Chloroplast
	*LsiNCED6*	Lsi09G011600	8.16	62.28	Chr09: 18638525 .. 18640216 (−)	1692	1692	563	−0.243	Cytoplasm
*Benincasa hispida*	*BhCCD1*	Bhi09G000536	6.16	61.37	Chr09: 13766074 .. 13779057 (−)	12984	1644	547	−0.266	Cytoplasm
	*BhCCD4*	Bhi01G002364	6.05	64.50	Chr01: 75445296 .. 75451135 (−)	5840	1764	587	−0.228	Chloroplast
	*BhCCD7*	Bhi12G001994	8.54	68.73	Chr12: 71046834 .. 71054528 (+)	7695	1830	609	−0.192	Chloroplast
	*BhCCD8*	Bhi04G000698	8.17	60.39	Chr04: 21509866 .. 21515484 (+)	5619	1632	543	−0.266	Chloroplast
	*BhCCDL*	Bhi09G002182	6.26	67.97	Chr09: 70181013 .. 70188430 (+)	7418	1785	594	−0.35	Chloroplast
	*BhNCED3*	Bhi01G002426	6.38	70.01	Chr01: 77692588 .. 77694885 (−)	2298	1902	633	−0.214	Chloroplast
	*BhNCED5*	Bhi03G000083	8.53	67.40	Chr03: 2257210 .. 2259286 (+)	2077	1803	600	−0.387	Chloroplast
	*BhNCED6*	Bhi12G001328	7.24	62.85	Chr12: 49163168 .. 49165223 (+)	2056	1707	568	−0.271	Chloroplast

## Data Availability

Not applicable.

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
