# Peer review of "Genome-Wide Identification of *CCD* Gene Family in Six Cucurbitaceae Species and Its Expression Profiles in Melon"

_genes, 2022, doi:10.3390/genes13020262_

Round 1

Reviewer 1 Report

The paper “Genome-wide identification of CCD gene family in Cucurbitaceae and its expression profiles in melon”by Denghu Cheng et al. report the identification of genes for carotenoid cleavage dioxygenase, the structure and expression of these genes and so on.

First of all, the figure quality is terrible and I could not read most part of the paper because I cannot understand the figures. Therefore, I could not evaluate the contents of the paper. I would like to ask the authors to make an overall revision of the paper.

  1. Figure 1-4: black background masks important letters.
  2. Figure 3-4: I cannot read any letters in the figure. There may be clade names in the left side of the figures. However, we cannot see anything.
  3. Figure 5-6: I cannot understand the heat map. What is the standard? Is the expression in root set as 0? We cannot see the expression level of these genes. So, in these experiments, the author should show the expression by bar graph.
  4. In the experiment of qRT-PCR, what gene was used for control?
  5. Table 1: Align text.

Author Response

Thank you for your kind suggestions.

1 We changed the picture again and added a background to the picture.

2. We changed the picture again, hoping to display the picture information clearly.

3 According to your suggestions, we changed the heat map to the bar chart.

We have added instructions to the "Materials and methods".

5. We have made serious adjustments to Table 1.

Reviewer 2 Report

  1. Please describe CCD gene functional information from Cucurbitaceae in the introduction part, As I know there are few reports showing the functional characterization of CCD genes in Cucurbitaceae.
  2. The introduction is poorly written
  3. What is the main aim of this study? Please describe the novelty and importance of the study and how this study is helping to address the carotenoid biosynthesis pathway in Cucurbitaceae.
  4. In lines 74 and 106, Please indicate the reference genome information. There are updated genome information’s for Melon, Cucumber, and Pumpkin.  
  5. From the genomic sequence comparison on the basis of the breeding history, resistance sources were introduced from okeechobeensis subsp.martinezii to C.moschata (Park et al 2020). Have you tried to analyze the CCD gene family information’s from such subspecies. I suggest authors include such subspecies and describe the source of CCD genes in Cucurbitaceae, this will give proper information on the evolution of the CCD gene family in Cucurbitaceae.
  6. Please improve the figure's quality and adjust them properly to the page. I suggest authors follow 1 column and 2 columns based on the width when making figures.
  7. Please cite and describe any experimental evidence of CCD genes corresponding to abiotic stress tolerance within Cucurbitaceae for the discussion.
  8.  Have you tried to analyze Apo carotenoids and compare them with CCD gene expression?     

Author Response

1 At your suggestion, I have described the functional information of CCD genes related to cucurbitaceae species in the introduction section

2 The introduction has been significantly revised and new references have been introduced

3 In the conclusion section, the main objectives of this study are introduced, and the novelty and importance of this study are described. Combined with the CCD gene of cucurbitaceae and related references, we believe that the CCD gene family will play an important role in the carotenoid biosynthesis pathway of cucurbitaceae through biotechnology, which is explained in the discussion section.

4 At your suggestion, we have included reference genomic information

5 I'm really sorry that there are very few studies on the CCD gene family of cucurbitaceae subspecies. Although we have consulted a large number of literatures, there are very few reports on the source of CCD gene of cucurbitaceae, but we will continue to carry out relevant studies according to your suggestions in future studies.

6 We reworked the images and improved the quality of the illustrations, adjusting them accordingly to the page

7We reviewed a large number of literature and in the discussion section illustrated experiments on CCD genes related to abiotic stress tolerance in cucurbitaceae.

8 I have reviewed a large number of literatures and found that the amount of apocarotenoid production is closely related to the amount of CCD gene expression. The description of this part is added in the preface.

Round 2

Reviewer 1 Report

The figures were improved. However, some figures should be improved.

  1. 3 and Fig. 4 should be improved. The resolution is very low and I cannot see the letters even when I magnify them. Figures and letters should be enlarged. Especially, legends on the upper-right and gene name.

(Please check the figures whether you can read the letter or not.)

Fig. 3C : yellow is very hard to see.

  1. I could not find the table of primers for qRT-PCR. Sequence of primers should be added.

  1. The authors sometime confuse genes and proteins.

L5-616: all CDD gene proteins cotain…

       It should be protein

L75: LmCCD1 cleaved

      It should be protein.

L121: protein sequence of all 9 CCD genes?

The authors should check the manuscript and revise these mistakes.

Usually, gene name should be written in Italic and protein name should be written in Roman.

Minor

  1. L64: it should be “carotenoid cleavage dioxygenase (CCD) pathway”.
